# The AFF4 scaffold binds human P-TEFb adjacent to HIV Tat

**Ursula Schulze-Gahmen[1], Heather Upton[1], Andrew Birnberg[1], Katherine Bao[1†a], Seemay Chou[1†b], Nevan J Krogan[2,3,4], Qiang Zhou[1], Tom Alber[1,3]\***

[1]Department of Molecular and Cell Biology, University of California, Berkeley, Berkeley, United States; [2]Department of Cellular and Molecular Pharmacology, University of California, San Francisco, San Francisco, United States; [3]California Institute for Quantitative Biosciences, QB3, Berkeley, United States; [4]J David Gladstone Institutes, San Francisco, United States

**Abstract** Human positive transcription elongation factor b (P-TEFb) phosphorylates RNA polymerase II and regulatory proteins to trigger elongation of many gene transcripts. The HIV-1 Tat protein selectively recruits P-TEFb as part of a super elongation complex (SEC) organized on a flexible AFF1 or AFF4 scaffold. To understand this specificity and determine if scaffold binding alters P-TEFb conformation, we determined the structure of a tripartite complex containing the recognition regions of P-TEFb and AFF4. AFF4 meanders over the surface of the P-TEFb cyclin T1 (CycT1) subunit but makes no stable contacts with the CDK9 kinase subunit. Interface mutations reduced CycT1 binding and AFF4-dependent transcription. AFF4 is positioned to make unexpected direct contacts with HIV Tat, and Tat enhances P-TEFb affinity for AFF4. These studies define the mechanism of scaffold recognition by P-TEFb and reveal an unanticipated intersubunit pocket on the AFF4 SEC that potentially represents a target for therapeutic intervention against HIV/AIDS.

**\*For correspondence:** tom@ucxray.berkeley.edu

**†Present address:** [a]Department of Immunology, Duke University School of Medicine, Durham, United States; [b]Department of Microbiology, University of Washington, Seattle, United States

**Competing interests:** The authors declare that no competing interests exist

**Reviewing editor**: Wes Sundquist, University of Utah, United States

## Introduction

At many genes in humans—including the integrated HIV genome as well as loci that regulate development and mediate responses to stress—RNA polymerase II initiates transcription but forms a stable paused complex after the synthesis of 30–50 nucleotides (*Lin et al., 2011*; *Levine, 2012*; *Luo et al., 2012a*; *Zhou et al., 2012*). These paused polymerases are poised for rapid, synchronous efficient transcription. For these genes, escape of the paused polymerase from the promoter-proximal region and elongation of the mRNA are rate-limiting regulated processes. Promoter escape requires positive transcription elongation factor b (P-TEFb), a heterodimeric protein kinase composed of CDK9 and cyclin T1 (CycT1) subunits. P-TEFb triggers promoter escape by directly or indirectly stimulating phosphorylation of the RNA polymerase II C-terminal domain and the associated factors, NELF (negative elongation factor) and DSIF (DRB sensitivity inducing factor). Consequently, recruitment of active P-TEFb to the paused polymerase complex serves as an important checkpoint for gene expression (*Levine, 2012*; *Luo et al., 2012b*; *Zhou et al., 2012*).

P-TEFb cycles between inactive and active complexes (*Zhou and Yik, 2006*). Recent studies of gene fusions in myeloid leukemias (*Lin et al., 2010*; *Yokoyama et al., 2010*), as well as complexes recruited to the HIV promoter by the HIV Tat protein (*He et al., 2010*; *Sobhian et al., 2010*; *Jager et al., 2012*), uncovered a family of Super Elongation Complexes (SECs) that bring together active P-TEFb and other transcription elongation factors. The SECs act at many normal human genes to stimulate mRNA elongation not only by triggering promoter escape but also by limiting proteolytic degradation of transcription elongation factors and increasing the processivity of RNAP II (*He et al., 2010*; *Lin et al., 2010*; *Biswas et al., 2011*; *Liu et al., 2012*). The SECs also couple to the PAF complex, which

**eLife digest** The rates at which many genes are expressed as proteins are limited by the efficiency of a process called transcriptional elongation. This process takes place as the stretch of DNA that defines the gene is transcribed into an RNA molecule and it is catalyzed by an enzyme called RNA polymerase II. However, this enzyme can become trapped, and another enzyme called P-TEFb (positive transcription elongation factor b) is needed to release it. P-TEFb and other elongation factors therefore have an important role in gene expression.

The human immunodeficiency virus (HIV) is a retrovirus that hijacks the gene expression processes in human immune cells to replicate the RNA genome of the virus. To do this, the virus produces a protein called Tat that recruits P-TEFb as part of a multi-protein machine called the super elongation complex. This ensures that the process of transcriptional elongation, and hence the overall replication process, is highly efficient. There are gaps, however, in our knowledge of the architecture of the super elongation complex, which is known to be organized on a flexible scaffold. In turn, the molecular basis for the interaction between HIV-1 Tat and P-TEFb within the super elongation complex is not well understood.

Now Schulze-Gahmen et al. show that only one of the two subunits in P-TEFb—a cyclin known as CycT1—binds to the AFF4 scaffold protein in the super elongation complex. In addition to assisting with the expression of hundreds of human genes, super elongation complexes containing P-TEFb-AFF4 are hijacked in various forms of cancer and viral infections, including HIV/AIDS. Schulze-Gahmen et al. show that AFF4 can directly contact HIV-1 Tat, which binds to the P-TEFb-AFF4 complex much more strongly than it binds to P-TEFb alone. This suggests that HIV-1 Tat evolved to work within the super elongation complex. Moreover, Schulze-Gahmen et al. reveal that HIV-1 Tat binds to a cleft between the P-TEFb enzyme and the AFF4 protein, which raises the possibility that this cleft could be used as a target for anti-HIV/AIDS drugs.

stimulates efficient transcript elongation (*He et al., 2011*). In addition to P-TEFb, the SECs contain subunits in the AF4, ELL, and ENL/AF9 protein families. Despite the major roles of SECs in metazoan gene expression and human disease, little is known about the architecture of these complexes.

To define the interactions that mediate SEC assembly and to understand how HIV Tat recruits P-TEFb (*Wei et al., 1998*) in the context of these large protein complexes, we mapped contacts among SEC subunits (*Figure 1A*) (*Chou et al., 2012*). Here, we report the structural and functional analysis of P-TEFb in complex with the cognate binding site on the SEC scaffold protein, AFF4. These studies reveal that AFF4 recognizes P-TEFb by binding the CycT1 subunit on the side opposite from CDK9. AFF4 is positioned to make direct contacts with HIV-1 Tat. Tat increases the affinity of P-TEFb for AFF4 by over an order of magnitude in vitro and rescues P-TEFb binding of AFF4 interface mutants in vivo. These results suggest that the SEC scaffold is an unanticipated direct partner of HIV-1 Tat, and an intersubunit Tat-binding pocket in the AFF4-P-TEFb complex may afford an unexpected site to target with selective inhibitors of HIV transcription.

## Results

The AF4 proteins, AFF1–4, form intrinsically disordered scaffolds that bind other transcription elongation factors not through folded domains but rather through dispersed short binding sites in the first 750 amino acids (*Chou et al., 2012*; *Leach et al., 2013*) (*Figure 1A*). Residues 2–73 of AFF4, for example, are sufficient to bind P-TEFb through the CycT1 subunit, and a peptide encompassing AFF4$_{2-73}$ folds upon binding CycT1. Using this binding site, we determined the 2.9-Å-resolution crystal structure of AFF4$_{2-73}$-P-TEFb-AMPPNP (R/R$_{free}$ = 0.207/0.245; *Figure 1*, *Figure 1—figure supplements 1 and 2*, and *Table 1*).

In all three independent copies of the complex in the crystals, AFF4 residues Leu34-Ile66 are ordered, binding to the second cyclin domain of CycT1 opposite CDK9 (*Figures 1B, 2*). In one complex, an isolated helix containing AFF4 residues 3–21 bridges symmetry-related CDK9 molecules in the crystals (*Figure 1—figure supplement 2*). We focus on the shared features of the three independent complexes. Consistent with the coupling of folding to binding, AFF4 lacks intramolecular tertiary contacts as it snakes across the CycT1 surface. An extended segment in AFF4 tracks

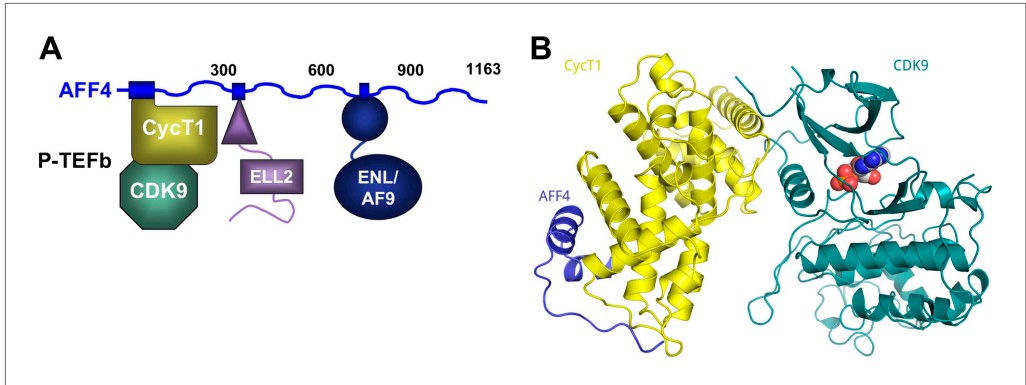

**Figure 1**. AFF4 binds CycT1 distal to CDK9. (**A**) Schematic model of the SEC. AFF4 is an intrinsically disordered scaffold that binds partners via 20–50 residue segments. (**B**) Ribbon diagram showing the strand–helix–helix arrangement of AFF4 (blue) bound to CycT1 (yellow) remote from CDK9 (teal). AFF4 adopts an extended conformation with no intramolecular tertiary contacts. AMPPNP (spheres) is bound to CDK9.

The following figure supplements are available for figure 1:

**Figure supplement 1**. Electron density for AFF4$_{2-73}$.

**Figure supplement 2**. A crystal contact formed by AFF4$_{2-21}$.

antiparallel to the H3'-H4' loop in CycT1, and two short AFF4 helices fit into shallow grooves in CycT1 between H3' and H5' and the surface formed by H2' and H3' (**Figure 2A**). Compared to the separated models, 1457 Å$^2$ of AFF4 and 1251 Å$^2$ of CycT1 accessible surface are buried in the complex. Twenty-six of the 33 ordered residues in AFF4 contact CycT1, emphasizing the extensive recognition determinants in the scaffold.

Many hydrophobic and aromatic residues in CycT1 mediate contacts with AFF4 (**Figure 2B–C** and **Figure 2—figure supplement 1**). A pocket between CycT1 Leu163-Val164-Arg165, Trp221, and Tyr224, for example, anchors AFF4 Phe35, which forms a classic edge-to-face interaction (**Burley and Petsko, 1985**) with the Tyr (**Figure 2C**). CycT1 Trp210 adjusts to create a site that buries AFF4 Pro38 and allows a short antiparallel β-sheet to form between AFF4 Tyr39-Val41 and CycT1 Asn 209-Glu211 (**Figure 2—figure supplement 1A**). CycT1 Trp207 shifts to form a hydrogen bond with the Gly57 carbonyl in the loop between the two AFF4 helices, additional van der Waals contacts with the loop backbone and nonpolar contacts with AFF4 residues Leu56 and Tyr59 (**Figure 2—figure supplement 1B**). Eight intermolecular hydrogen bonds help establish the chemical complementarity between AFF4 and CycT1. Twenty-one of the 26 interacting residues in AFF4 are conserved in AFF1–3 (**Figure 2—figure supplement 2**).

In comparison to the structure of P-TEFb alone (CDK9$_{1-330}$/CycT1$_{1-259}$; PDB ID 3BLQ; **Baumli et al., 2008**), the last 20 residues present in the CycT1 subunit undergo a major rearrangement upon AFF4 binding (**Figure 2D**). AFF4 overlaps with the position of residues 243–259 in the superimposed P-TEFb cyclin subunit. To accommodate AFF4, helix H5' straightens and residues 251–256 form a helical segment that packs against CycT1 helices H1 and H2' and forms part of the AFF4 binding surface. In addition to burying hydrophobic residues in the first AFF4 helix, the adjustments in CycT1 helix H5' mediate formation of a hydrogen-bonded ion pair between AFF4 Arg51 and CycT1 Glu246. CycT1 Trp256, the last residue ordered in the AFF4 complex, moves over 13 Å upon AFF4 binding.

To probe the basis for CycT1 binding, we measured the effects of interface mutations on the stability of the AFF4 complex in vitro. The AFF4 2–363 and 33–67 fragments bound to CycT1 with similar affinities (**Table 2**), suggesting that the ordered AFF4 segment in the crystal structure captures the major CycT1 binding determinants. Mutations throughout the CycT1 interaction surface reduced AFF4 binding (**Figure 3A** and **Table 3**). The Trp210Ala and Trp207Ala substitutions in CycT1 had the largest effects, respectively, reducing AFF4 affinity by 21- and 58-fold. These results point to these CycT1 Trp residues as interaction hotspots and suggest that contacts all along the interface observed in the crystal structure mediate AFF4 recognition.

**Table 1.** X-ray data collection and refinement statistics for AFF4-P-TEFb-AMPPNP

| Data collection | AFF4-P-TEFb-AMPPNP |
| --- | --- |
| Space group | $P2_12_12_1$ |
| Cell dimensions: *a, b, c* | 100.691, 126.298, 195.626 |
| Resolution (Å)* | 50-2.94 (2.99–2.94) |
| Unique reflections* | 54,189 (2664) |
| $R_{sym}$ (%)* | 9.3 (>100) |
| $I/\sigma(I)$* | 23.2 (1.3) |
| Completeness (%)* | 100.0 (100.0) |
| Redundancy* | 8.1 (7.5) |
| Temperature (K) | 100 |
| Mosaicity (°) | 0.45–0.6 |
| **Refinement** | |
| Resolution (Å) | 48.7-2.94 |
| No. reflections | 53,775 |
| $R_{work}/R_{free}$ | 0.207/0.245 |
| No. atoms/B-factors ($Å^2$) | |
| CDK9, molecule 1, 2, 3 | 2558 (111.9), 2533 (116.3), 2558 (121.6) |
| Cyclin T1, molecule 1, 2, 3 | 2003 (121.3), 2024 (123.1), 2001 (118.5) |
| AFF4, molecule 1, 2, 3 | 248 (156.3), 421 (161.1), 243 (160.3) |
| Water | 19 (90.1) |
| Root mean square deviations | |
| Bond lengths (Å) | 0.004 |
| Bond angles (°) | 0.666 |
| Ramachandran plot† | |
| Favored (%) | 94.7 |
| Allowed (%) | 4.48 |
| Disallowed (%) | 0.78 |
| Protein Data Bank ID | 4IMY |

*Values in parentheses are for the highest resolution shell.
†Values from MOLPROBITY.

To examine the functional roles of the N-terminal 72 residues of AFF4, we measured the effects of tandem alanine mutations on the stimulation of expression of a luciferase reporter gene driven by the HIV-1 promoter in HeLa cells (*He et al., 2011*) (*Figure 3B* and *Figure 3—figure supplement 1*). Consistent with the structure, mutations of AFF4 residues that contact P-TEFb reduced Tat-independent transcriptional stimulation. Three of the most deleterious variations—alanine substitutions at Pro33/ Leu34, Phe35 (Ala36 was maintained), and Met55/Leu56—shorten hydrophobic side chains that are buried in the interface. Tandem alanine substitutions (Glu45/Asp46 and Gly57/Asn58) that remove side chains that cap and stabilize the AFF4 helices also reduced luciferase expression. In contrast, residues such as Glu61/Met62 and Ile71/Pro72, which are more tolerant of di-alanine substitutions, are more solvent exposed or flexible. Tandem alanine replacements in residues 3–32, which flank the ordered CycT1 contacts in the crystal structure, showed significant but generally smaller effects on AFF4 transcriptional stimulation activity (*Figure 3B*). These results implicate the P-TEFb binding site, as well as the flanking flexible sequences, in the function of the AFF4 N-terminal segment.

The HIV-1 Tat protein was shown nearly 15 years ago to bridge P-TEFb and the TAR RNA element near the 5' end of nascent HIV transcripts to recruit the active CDK9 kinase to the HIV promoter (*Wei et al., 1998*). The discovery that Tat recruits P-TEFb as part of a larger SEC (*He et al., 2010*; *Sobhian et al., 2010*) raised the question of how Tat distinguishes the SEC from free P-TEFb, particularly since

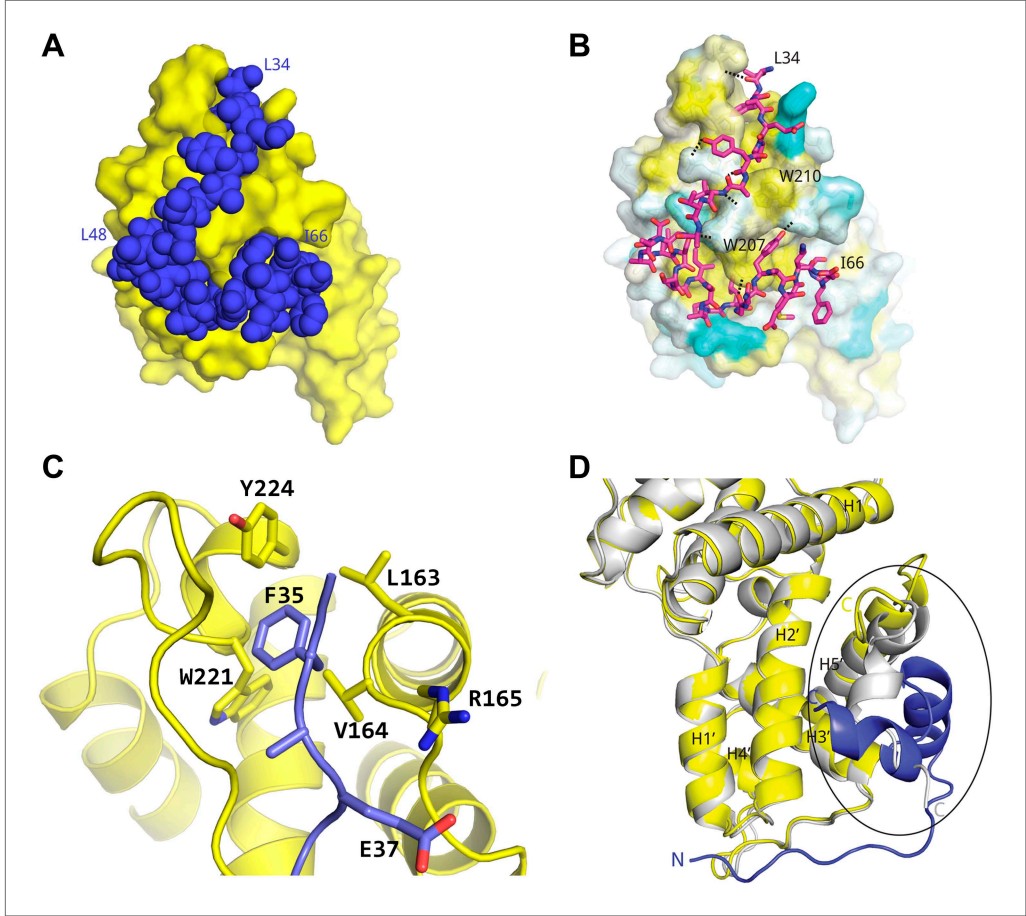

**Figure 2**. Basis for AFF4 scaffold recognition by P-TEFb. (**A**) AFF4 residues 34–66 (blue spheres) fill grooves on CycT1 (yellow surface). (**B**) Chemical complementarity mediates AFF4 binding. Exposed hydrophobic residues of CycT1 (yellow surface) are buried in the AFF4 complex. Hydrogen bonds (black dotted lines) also mediate binding. (**C**) AFF4 Phe35 is buried in a hydrophobic pocket formed by aromatic and nonpolar residues on the surface of CycT1. (**D**) The C-terminus of the CycT1 cyclin domain (gray in the ellipse) adjusts to make contacts with AFF4 (blue).

The following figure supplements are available for figure 2:

**Figure supplement 1**. Example interactions between AFF4 and P-TEFb.

**Figure supplement 2**. Conserved AFF4 sequences mediate P-TEFb recognition.

Tat binds P-TEFb in isolation. Moreover, Tat shows specificity for SECs containing AFF4 and AFF1 (*He et al., 2010*; *Sobhian et al., 2010*). What accounts for this specificity? The AFF4-P-TEFb crystal structure provides a simple and unsuspected explanation—Tat binds in a position to make direct contacts with the scaffold (*Figure 4A*). Superposition of the CycT1 subunits of the structures of Tat-P-TEFb (*Tahirov et al., 2010*) and AFF4-P-TEFb shows that Tat is positioned to pack against helix 2 of AFF4. Tat Met1, Lys28 (which is reversibly acetylated in vivo; *Kiernan et al., 1999*; *Ott et al., 2011*) and Phe32, in particular, are predicted to interact with AFF4 Glu61, Met62, Phe65, and Ile66. The disordered C-terminus of the AFF4 peptide also neighbors Glu2 and His13-Gly15 of Tat. These seven residues of Tat are crucial for transcriptional activation, even though they are exposed to solvent in the Tat-P-TEFb complex (*D'Orso et al., 2012*).

The juxtaposition of AFF4 and Tat on the P-TEFb surface predicts that the scaffold enhances Tat binding. Direct measurements of Tat affinity are problematic, however, because Tat is unstable in vitro in the absence of partners and difficult to maintain in an active form. To overcome this problem, we took advantage of a thermodynamic cycle that illustrates that AFF4 and Tat mutually influence the P-TEFb affinity of each other by the same amount (*Figure 4—figure supplement 1*). In quantitative in vitro

**Table 2.** Binding affinities of AFF4 segments

| | Direct binding | Competition assay | | |
| --- | --- | --- | --- | --- |
| | AFF4$_{32-67}$ | AFF4$_{32-67}$ | AFF4$_{2-73}$ | AFF4$_{2-363}$ |
| CycT1 | 104 ± 17 nM | 102 ± 10 nM | 130 ± 18 nM | 115 ± 15 nM |
| P-TEFb | 36 ± 6 nM | 36 ± 4 nM | 10 ± 1 nM | 7 ± 1 nM |
| Tat-P-TEFb | 8.8 ± 0.8 nM | 4.5 ± 0.6 nM | 0.85 ± 0.15 nM | 0.6 ± 0.1 nM |

Dissociation constants measured by direct binding of fluorescein-labeled AFF4$_{32-67}$ and by competition with unlabeled AFF4 segments. The increased affinity of AFF4 for P-TEFb compared to CycT1 may be due to structural changes in the cyclin subunit or additional interactions with the CDK9 kinase subunit. The similar affinities of AFF4$_{2-73}$ and AFF4$_{2-363}$ for all the cyclin-containing species suggest that AFF4$_{2-73}$ encompasses the binding sites for P-TEFb and Tat-P-TEFb.

assays (*Figure 4B* and *Table 2*), the purified AFF4$_{2-73}$ peptide bound Tat-P-TEFb (K$_d$ = 0.85 ± 0.15 nM) ~11 times more tightly than P-TEFb (K$_d$ = 10 ± 1 nM). The AFF4$_{2-363}$ segment also bound Tat-P-TEFb (K$_d$ = 0.6 ± 0.1 nM) ~11 times more tightly than P-TEFb (K$_d$ = 7 ± 1 nM). These results support the model of direct Tat interactions with AFF4 and suggest that the AFF4$_{2-73}$ peptide includes the principal residues that contact Tat (*Figure 4* and *Figure 4—figure supplement 1*).

To test the importance of the AFF4 interactions for P-TEFb and Tat recognition in vivo, we measured the effects of mutations in AFF4 on the binding of SEC subunits and Tat in HeLa cells. Cells were transfected with wild-type or mutant AFF4 containing a C-terminal 3× FLAG tag, and immunoprecipitations using an anti-FLAG antibody were probed for the presence of associated factors. Alanine substitutions in P-TEFb contacts such as AFF4 Phe35, Tyr59/Asp60, and Lys63/Asp64 reduced binding of P-TEFb but not the ELL2, AF9, or ENL subunits of the SEC (*Figure 4C*). These defects were rescued by overexpressing a stably integrated gene for Tat, which strengthened the AFF4-P-TEFb association. In contrast, tandem alanine substitutions for AFF4 Glu61/Met62, which are more exposed in the P-TEFb interface and predicted to contact Tat, caused a small reduction (~30%) in the binding of P-TEFb that was not rescued by overexpressing Tat (*Figure 4C*). The specificity of these mutational effects supports the model that Tat interacts directly with AFF4.

In keeping with the tenuous contacts of the AFF4$_{3-21}$ helix with CDK9, AFF4$_{2-73}$ (K$_d$ = 10 ± 1 nM) binds only 3.6 times more tightly than AFF4$_{32-67}$ (K$_d$ = 36 ± 4 nM) to P-TEFb (*Table 2*). In addition, the CDK9 kinase subunit structure is little changed upon AFF4 binding to P-TEFb (CDK9$_{15-360}$ RMSD = 0.54 Å vs 1BLQ). In vivo, the Glu13Ala/Arg14Ala AFF4 mutant in the heart of the CDK9 interface in the crystals is associated with a <25% decrease in transcription stimulation (*Figure 3B*) and little change in P-TEFb binding (*Figure 4C*). To measure if recruitment of P-TEFb to the SEC scaffold regulates the kinase, we assayed the effects of AFF4 on the in vitro phosphorylation of an RNA polymerase II CTD substrate by purified P-TEFb and Tat-P-TEFb (*Bitoun et al., 2007*). AFF4$_{2-73}$ inhibited CTD phosphorylation by approximately twofold in a purified system (*Figure 5* and *Figure 5—figure supplements 1 and 2*). By comparison, Tat stimulated P-TEFb by sevenfold, and addition of AFF$_{2-73}$ did not further stimulate the kinase activity of the Tat-P-TEFb complex. Taken together, these results point away from CDK9 as the primary physiological partner of AFF4$_{3-21}$ and suggest that AFF4 functions as a SEC scaffold but not an allosteric activator of CDK9.

## Discussion

The AFF4-P-TEFb crystal structure reveals that a high density of contacts in residues 34–66 of the AFF4 scaffold mediates binding to the CycT1 subunit of P-TEFb. These contacts are largely conserved in AF4 family members and can be perturbed physically and functionally by mutations (*Figure 3* and *Figure 2—figure supplement 2*). Consistent with these results, tandem alanine substitutions of Pro33/Leu34, Val41/Thr42, Arg51/Ile52, and Met55/Leu56 in the CycT1 binding site (but not Arg3/Glu4 or Glu25/Asp26 in the preceding segment) also reduce P-TEFb binding in vivo (*Chou et al., 2012*). The impacts of alanine substitutions on transcriptional stimulation by AFF4 (*Figure 3B*) show that residues in the P-TEFb interface, as well as helix stabilizing residues, play crucial roles. The additional sensitivity of transcriptional stimulation to alanine substitutions of disordered residues flanking the CycT1 binding site suggests that the flexibility and potential interactions with other ligands are important for function.

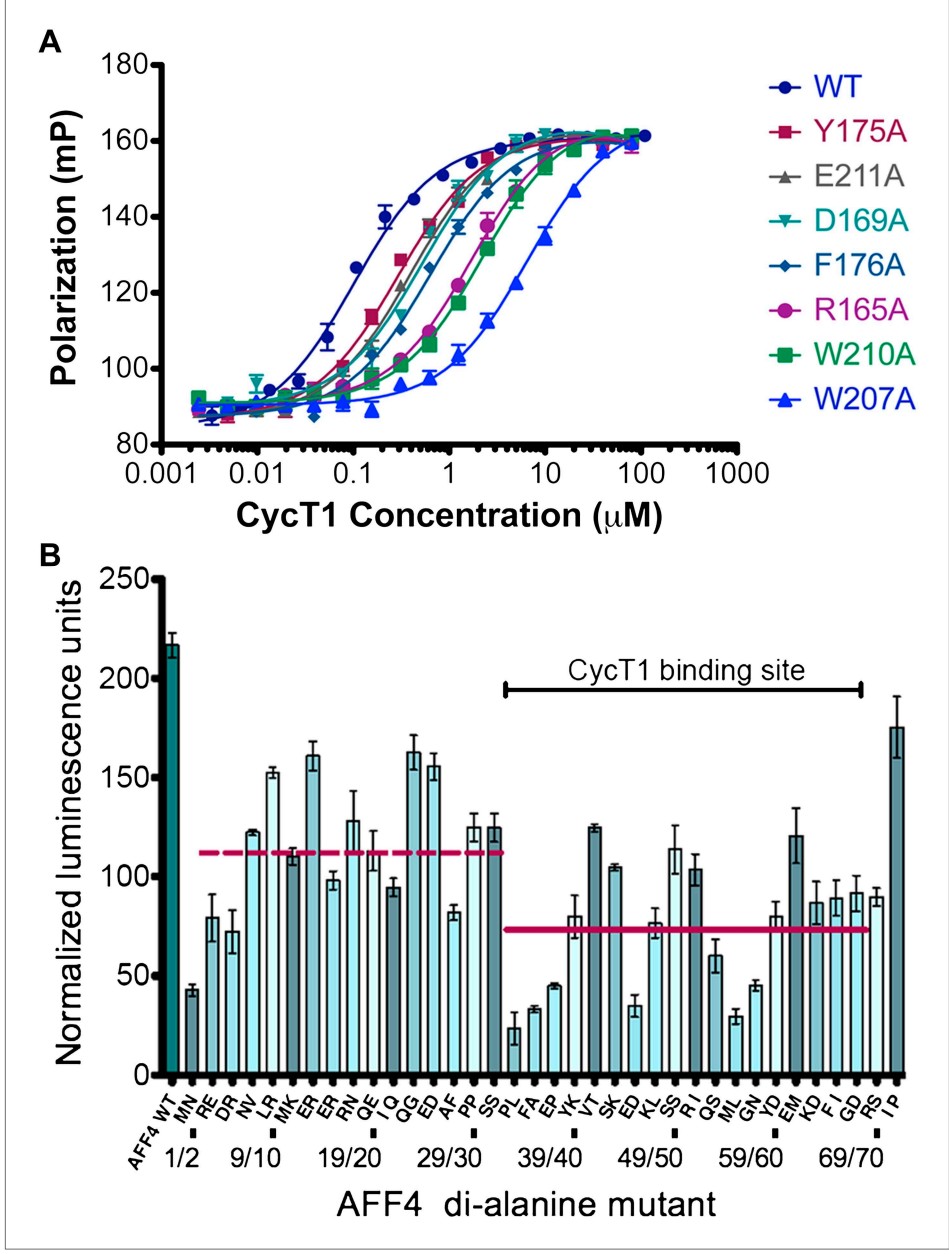

**Figure 3**. AFF4 interface mediates P-TEFb recognition. (**A**) Mutations of CycT1 contact residues reduce AFF4 affinity. Fluorescence polarization of fluorescein-labeled AFF4$_{32-67}$ (5 nM) is plotted as a function of the concentration of the indicated CycT1 variant. (**B**) Transcriptional effects of AFF4 tandem Ala mutants. Stimulation of Tat-independent transcription from the HIV LTR was measured in extracts of cells cotransfected with a luciferase reporter construct and an expression vector for the indicated di-Ala AFF4 variant. Activity was normalized to the level of AFF4 expression. Values represent the mean of three independent assays. Tandem alanine substitutions cover the first 72 residues of AFF4. Horizontal lines correspond to the mean stimulation of di-Ala substitutions in residues 3–32 (left; 113.9 ± 5.1) and 33–66 (right; 73.6 ± 4.9).

The following figure supplements are available for figure 3:

**Figure supplement 1**. Expression levels of AFF4 variants.

HIV-1 Tat binds adjacent to AFF4, increases the affinity of AFF4 for P-TEFb by over an order of magnitude, and rescues P-TEFb binding to AFF4 interface mutants in vivo. The AFF4 Glu61/Met62 double alanine substitution in the proposed Tat interface blocks this rescue. These results suggest that

**Table 3.** Dissociation constants of AFF4$_{32-67}$ for Cyclin T1 mutants

| CycT1 variant | Kd (nM) |
|---|---|
| Wild-type | 104 ± 17 |
| Y175A | 228 ± 18 |
| E211A | 356 ± 29 |
| D169A | 438 ± 41 |
| F176A | 645 ± 58 |
| R165A | 1592 ± 171 |
| W210A | 2190 ± 246 |
| W207A | 6050 ± 871 |

direct contacts between Tat and the AFF4 scaffold in complex with P-TEFb mediate the selective recruitment of the SEC to the HIV promoter. The functional AFF4-Tat interface, including the acetylated Lys28 in Tat as a potential regulator of affinity (*Kiernan et al., 1999*), supports the idea that Tat evolved to function within the SEC. In the crystal structure of the (low-affinity) Tat-P-TEFb subcomplex (*Tahirov et al., 2010*), Tat binds to a relatively open groove. In the context of the AFF4-P-TEFb structure, however, AFF4 creates an unanticipated pocket for Tat (*Figure 4D*). This pocket may ultimately provide a suitable therapeutic target for the development of small-molecule inhibitors of Tat binding that selectively block HIV transcription.

## Materials and methods

### Expression of P-TEFb, P-TEFb-Tat, and AFF4

CDK9 (1–330) and cyclin T1 (1–264) were cloned into a modified pFastBac Dual donor plasmid, and HIV-1 Tat (1–86) was cloned into the pFastBac1 donor plasmid using the Bac-To-Bac system from Life Technologies (Carlsbad, CA). CDK9 was cloned with a Tobacco Etch Virus (TEV) protease cleavable N-terminal His-tag while CycT1 and Tat remained un-tagged. Each virus genome was transfected into Sf9 cells to generate the baculovirus according to manufacturer's protocol. Each baculovirus was amplified (2×), plaque purified, and amplified (3×) to obtain a stock with 10$^8$ plaque forming units (PFU)/ml. Test infections were screened for expression levels of the target proteins by Western blot analysis, and the highest expressing virus stock was used.

For large-scale production of P-TEFb, 4 l of High5 cells at 1 × 10$^6$ cells/ml in ESF921 medium (Expression Systems, Reno, NV) were infected with 20 ml of virus stock per liter of culture. The flasks were incubated for 52 hr at 27°C on a rotary shaker. Cells were harvested by centrifugation for 30 min at 350x$g$ in a Beckmann JLA-8.1 rotor, washed quickly in 50 mM Tris pH 7.5, 150 mM NaCl, and centrifuged at 350×$g$ for 10 min. The supernatant was removed, and the cell pellets were frozen in liquid nitrogen. For large-scale production of P-TEFb-Tat, insect cells were coinfected with 20 ml each of CDK9, CycT1, and Tat virus stocks. The remaining steps were the same as for production of P-TEFb.

AFF4$_{2-73}$ was cloned into a modified pET28 plasmid. The recombinant protein includes a N-terminal TEV-protease-cleavable His-tag. The plasmid was transformed into Rosetta(DE3)pLysS. 3 l of transformed *Escherichia coli* were grown at 37°C to OD$_{600}$ = 0.6, and expression was induced with 0.3 mM IPTG at 18°C for 16 hr. Cells were harvested, and pellets were frozen in liquid nitrogen.

### Purification of the AFF4-P-TEFb complex

Pellets from 4 l infected High5 cells were resuspended in 75 ml lysis buffer (20 mM Na-HEPES pH 7.4, 10 mM NaCl, 1 mM DTT) with 1 × Roche Complete Protease Inhibitor and 0.5 mM AEBSF. Cells were lysed with a Dounce homogenizer. The lysate was brought up to 0.2 M NaCl by adding 3.0 ml 5 M NaCl, incubated on ice for 10 min, and centrifuged at 5800x$g$ in a SS34 rotor. The supernatant was saved, and the pellet extracted again with 30 ml lysis buffer in the homogenizer. The supernatants of the two centrifugations were combined, cleared by centrifugation at 210,000×$g$ in an ultracentrifuge, and filtered through a 0.8-µm syringe filter. The cleared lysate was loaded onto a 5 ml His-Trap HP column (GE Healthcare, Piscataway, NJ) equilibrated in buffer A (20 mM Na-HEPES pH 7.4, 0.3 M NaCl, 10% glycerol, 1 mM DTT, 20 mM imidazole). After washing for 10 column volumes with buffer A + 1 M NaCl, followed by 10 column volumes of buffer A, P-TEFb was eluted with a gradient from 0% A to 100% B (20 mM Na-HEPES pH 7.4, 0.3 M NaCl, 10% glycerol, 0.5 M imidazole, 1 mM DTT). The eluted P-TEFb was dialyzed for 3 hr against 2 L buffer A. TEV-protease was added to the protein at a 1:25 (wt/wt) ratio, and the digest was incubated for 1 hr at room temperature and 4°C overnight. The digest was loaded on a His-Trap HP column, and P-TEFb lacking the His tag eluted in the flow-through of the

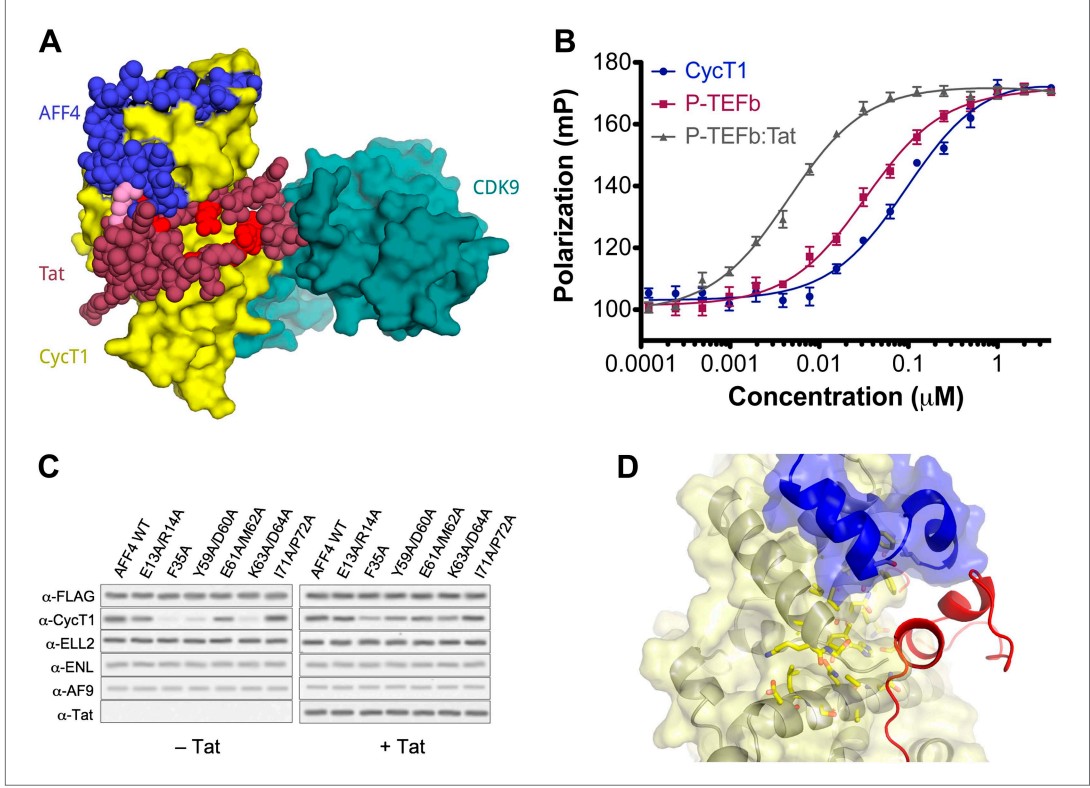

**Figure 4**. AFF4 binds in position to make direct contacts with HIV-1 Tat. (**A**) Superposition of the AFF4-P-TEFb complex and the Tat-P-TEFb complex using the cyclin subunit (yellow) shows the close proximity of AFF4 (blue) and Tat (red). Tat Lys28 (pink), where acetylation stimulates function, as well as other residues essential for Tat transcriptional activation (**D'Orso et al., 2012**) that are exposed to solvent in the Tat-P-TEFb complex (bright red) are positioned adjacent to AFF4. (**B**) Tat enhances AFF4 binding in vitro. Fluorescence polarization of fluorescein-labeled AFF4$_{32-67}$ (5 nM) is plotted as a function of the concentration of CycT1 (blue circles), P-TEFb (red squares), and Tat-P-TEFb (green triangles). (**C**) Alanine substitutions in the P-TEFb binding site of AFF4 reduce CycT1 binding but not associations of other SEC subunits in HeLa cells. Western blots show associations of each indicated factor with different FLAG-tagged AFF4 variants (top). Lysates were immunoprecipitated with an anti-FLAG antibody. Expression of Tat (right) rescues defects in CycT1 binding, except for the E61/M62 double alanine mutant. This mutant in the predicted AFF4-Tat interface shows equal small defects in P-TEFb binding in the absence (left) and presence (right) of Tat. (**D**) AFF4 (blue) and CycT1 (yellow) create an intersubunit pocket where Tat (red) can bind with minor structural adjustments. The program DoGSiteScorer (**Volkamer et al., 2012**) assigns this cleft a high druggability score (0.83 out of 0–1.0) and shows that it contains the most nonpolar surface of any pocket in the AFF4-P-TEFb structure.

The following figure supplements are available for figure 4:

**Figure supplement 1**. Thermodynamic cycle for AFF4 and Tat binding to P-TEFb.

column, while undigested protein and TEV-protease eluted later in the imidazole gradient. The yield was ~0.8 mg P-TEFb/L High5 cell culture.

The Tat-P-TEFb complex was purified in a similar way. The cell lysate was purified over a 5 ml His-Trap HP column, dialyzed, and digested with TEV protease as described above. The digested complex was diluted with 1.1 volumes of 20 mM Na-HEPES pH 7.3, 1 mM DTT, 1% β-octyl-glucoside to a final concentration of 0.14 M NaCl, and 0.5% β-octyl-glucoside and applied to a Source S anion exchange column equilibrated in 20 mM Na-HEPES pH 7.3, 10% glycerol, and 1 mM DTT. The column was developed with a linear gradient to 20 mM Na-HEPES pH 7.3, 0.75 M NaCl, 10% glycerol, and 1 mM DTT. P-TEFb-Tat eluted as single peak at about 0.25 M NaCl.

To purify AFF4$_{2-73}$, 30 g of *E. coli* cell pellet was resuspended in 125 ml lysis buffer (25 mM Tris/HCl pH 7.5, 0.2 M NaCl, and 1 mM DTT). Lysozyme was added to 0.1 mg/ml final concentration and incubated for 30 min. After adding Roche Complete Protease Inhibitor without EDTA, 0.5 mM AEBSF, and

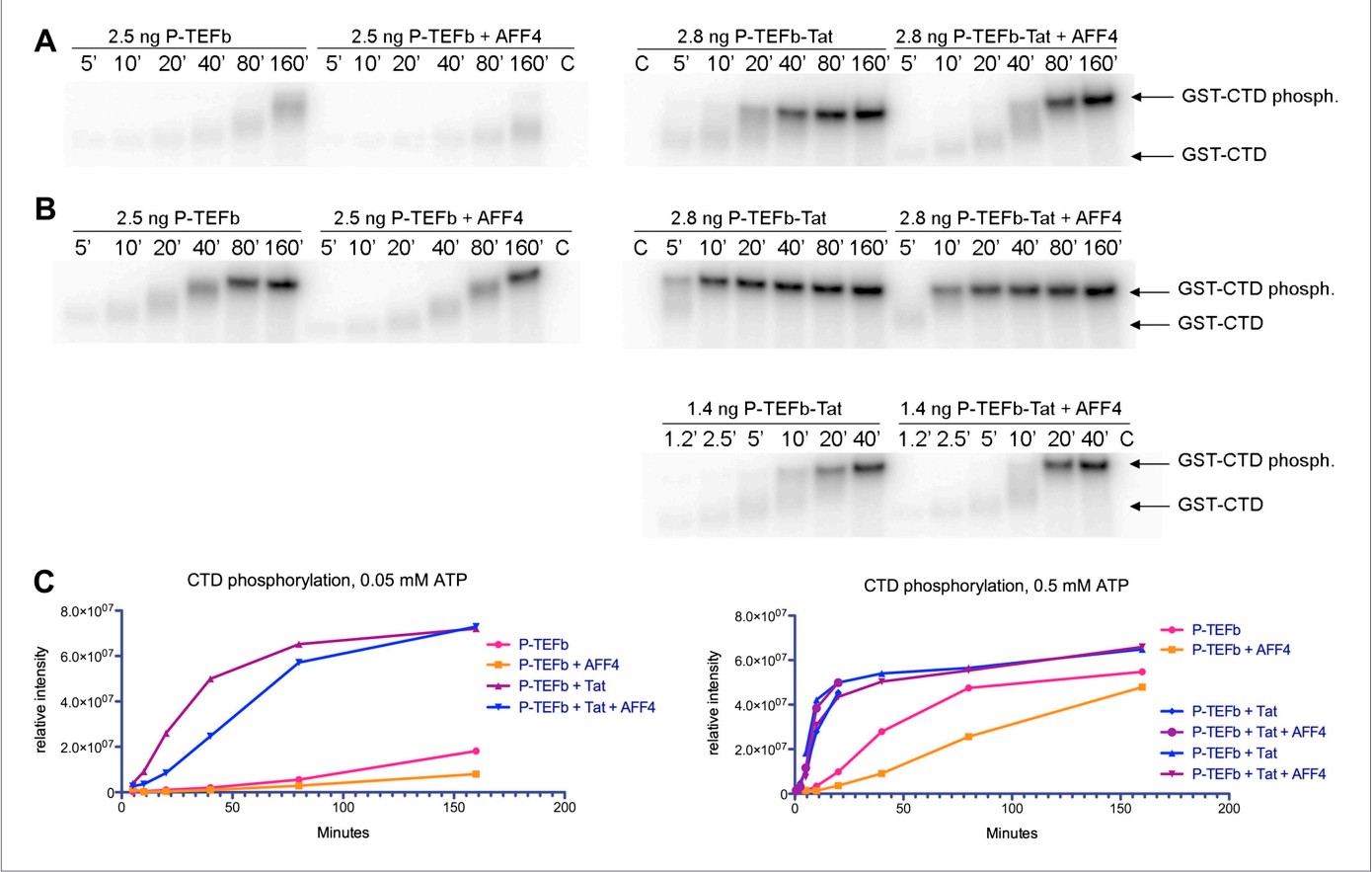

**Figure 5**. Kinase activity of P-TEFb and P-TEFb-Tat complexes with AFF4. (**A**) Autoradiogram showing phosphorylation of GST-CTD (500 ng) by P-TEFb and P-TEFb-Tat with and without excess (0.28 µM) AFF4$_{2-73}$ in the presence of low (50 µM) ATP. (**B**) Phosphorylation of 500 ng GST-CTD by P-TEFb and P-TEFb-Tat with and without excess (0.28 µM) AFF4$_{2-73}$ in the presence of saturating (500 µM) ATP. AFF4 reduces the activity of P-TEFb twofold and has little influence on the kinase activity of Tat-P-TEFb. Tat-P-TEFb is, however, sevenfold to tenfold more active than P-TEFb. Lane 3 in panels (**A** and **B**) is a control without GST-CTD. (**C**) Quantitation of the radioactive GST-CTD in (**A** and **B**).
The following figure supplements are available for figure 5:

**Figure supplement 1**. SDS polyacrylamide gel of P-TEFb and P-TEFb-Tat at the same ratio as they were used in the kinase assay.

**Figure supplement 2**. Western blots of kinase reaction products from panel B. Phosphorylated CTD was detected with anti-phoshoSer2 and anti-phoshoSer5 antibodies.

DNaseI (5 units/ml), the cells were lysed by sonication. The lysate was centrifuged for 1 hr at 17,000 rpm in a SS34 rotor. The supernatant was filtered through a 0.8-µm syringe filter and applied to a 5 ml His-Trap column. The protein was purified as described for P-TEFb.

The separately purified P-TEFb and AFF4$_{2-73}$ were combined at a 1:1.4 (mol/mol) ratio, concentrated to 0.6 ml, and injected onto an analytical Superdex S200 gel exclusion column equilibrated with 25 mM Na-HEPES pH 7.4, 0.2 M NaCl, and 1 mM DTT. The center fractions of the eluted three-protein peak were used for crystallization.

## Crystallization and structure determination

The purified AFF4-P-TEFb complex was concentrated to 10 mg/ml using Millipore Ultrafree centrifugal devices. Crystals were grown from 1.0 µl protein combined with 0.5 µl silver bullet condition 30 (Hampton Research, Aliso Viejo, CA; 0.33% wt/vol Gly-Phe, 0.33% wt/vol Gly-Tyr, and 0.33% wt/vol Leu-Gly-Gly in 20 mM Na-HEPES pH 6.8) and 0.5 µl reservoir solution equilibrated against 16–18% PEG 3350, 0.1 M Na-HEPES pH 7.0. After equilibrating for 24 hr, diluted microseeds from previous

crystallization experiments were added with a hair. Seeding produced large single crystals (0.3 × 0.2 × 0.2 mm) with and without 1 mM AMPPNP, 5 mM $MgCl_2$. The presence of all three proteins in the crystals was confirmed by gel electrophoresis and mass spectrometry of dissolved crystals.

Crystals were soaked in cryoprotectant (25% PEG 3350, 0.1 M Na-HEPES pH 7.0, 10% glycerol, 1:4 silver bullet 30) and flash frozen in liquid nitrogen. X-ray data were collected at Beamline 8.3.1 at the Advanced Light Source at the Lawrence Berkeley National Laboratory (*MacDowell et al., 2004*). The best data were collected from a crystal that was grown in the presence of 1 mM AMPPNP and 5 mM $MgCl_2$. The reflections were processed using HKL2000 (*Otwinowski and Minor, 1997*) (*Table 1*).

The structure was determined by molecular replacement with PHENIX (*Adams et al., 2010*) using P-TEFb from the P-TEFb-Tat complex (PDB ID 3MI9) as the search model. The asymmetric unit contains three complexes. Initial refinement using PHENIX was performed with model restraints, as well as noncrystallographic symmetry restraints. Model restraints were removed in later stages of refinement. AFF4 was built manually and the model was adjusted using Coot (*Emsley and Cowtan, 2004*). The model was refined using gradient minimization with weight optimization and maximum-likelihood targets, TLS refinement, and individual atomic B-factor refinement. The model was checked against composite omit maps. Density was missing for residues 1–7 and 88–95 in CDK9 mol1 and mol3, and residues 1–7 and 89–97 in CDK9 mol2. Density also was absent for residues 1–7 and 253–264 in CycT1 mol1, residues 1–7 and 257–264 in CycT1 mol2 and mol3, and residues 2–33 and 67–73 in AFF4, mol1 and mol3. For AFF4 in molecule 2 residues 2, 22–33, and 67–73 are missing.

The register of the AFF4 sequence in the electron density was confirmed by locating the Se atoms in crystals grown with SeMet-labeled AFF4. $AFF4_{2–73}$ was labeled with SeMet (*Van Duyne et al., 1993*) and purified as described above. Optimized crystallization conditions were seeded with unlabeled microcrystals. Anomalous differences were calculated from data collected at 12,657 eV, the peak of Se fluorescence measured from the crystal.

## Structure analysis

Buried surface areas were calculated with PISA (*Krissinel and Henrick, 2007*). Surface pockets were identified with the program DoGSiteScorer (*Volkamer et al., 2012*). Figures of molecular structures were prepared with PyMOL Version 1.5 (Schroedinger LLC, New York, NY).

## AFF4 affinity for CycT1

Protein binding was measured using fluorescence anisotropy of a 36-residue segment of AFF4 (residues 32–67) encompassing the protein–protein contacts in the crystal structure. The AFF4 peptide was synthesized at the University of Utah DNA/Peptide Facility using the following sequence: C-FAM-GABA-SPLFAEPYKVTSKEDKLSSRIQSMLGNYDEMKDFIG-amide where FAM indicates 5-carboxyfluoroscein and GABA indicates a γ-amino-butyric acid spacer. Varying amounts of purified CycT1, P-TEFb, and Tat-P-TEFb were incubated for 30 min with 5 nM labeled peptide at room temperature in the dark in 25 mM HEPES pH 7.5, 100 mM NaCl, 10% glycerol, 0.1% NP40, and 0.5 mM Tris(2-carboxyethyl)phosphine (TCEP). Competition titration experiments of unlabeled peptides were performed using 75 nM CycT1, 45 nM P-TEFb, and 6 nM Tat-P-TEFb in 25 mM HEPES pH 7.5, 100 mM NaCl, 10% glycerol, 0.1% NP40, 0.5 mM TCEP, and 5 nM fluorescent peptide. Fluorescence anisotropy was measured using a Victor 3V (Perkin Elmer) multi-label plate reader. Data points represent the average of six independent measurements. Binding curves were fit to the single-site binding equation using Prism version 5.0c (Graphpad Software).

## AFF4 stimulation of Tat-independent transcription

HeLa cells were cultured in Dulbecco's modified Eagle's medium supplemented with 10% FBS at 37°C in a humidified atmosphere with 5% $CO_2$. Cells were seeded at 5 × $10^5$ cells/ml in TC-treated 96-well plates one day prior to plasmid transfection using the 25-kDa linear polyethyleneimine reagent (Sigma-Aldrich, St. Louis, MO). Cells were cotransfected with 100 ng of an HIV-LTR firefly luciferase reporter construct (*He et al., 2010*) and 350 ng of pCDNA3.1 containing AFF4 variants with a C-terminal 3× FLAG tag. Following stimulation for 48 hr with the indicated ligands, the cells were lysed in passive lysis buffer (Promega, Fitchburg, WI) for 5 min at 25°C. The cell lysates were incubated with firefly luciferase substrate, and luminescence was measured on a SpectraMax L microplate reader (Molecular Devices, Sunnyvale, CA). The relative luminescence was normalized to the concentration of AFF4 in the cell determined by Western blotting using an anti-FLAG primary antibody.

## Co-immunoprecipitation and detection of proteins bound to AFF4

HeLa cells were seeded in 10-cm TC-treated plates at $2.2 \times 10^5$ cells/ml, Incubated for 24 hr and transfected with pCDNA3.1 (1 µg) encoding an AFF4 variant with a C-terminal 3× FLAG tag. After incubating for 2 days, cells were collected in PBS, washed, and resuspended in hypotonic buffer (20 mM Tris-HCl pH 7.4, 10 mM NaCl, and 3 mM $MgCl_2$). After 15 min on ice, 0.4% Triton X-100 was added, and the cell suspension was mixed and centrifuged at 3000×$g$ for 10 min. The pellet was resuspended in nuclear extraction buffer (100 mM Tris-HCl pH 7.4, 100 mM NaCl, 1% Triton X-100, 1 mM EDTA, 10% glycerol, 0.7% Tween 20, and protease inhibitors [AEBSF, leupeptin and E64]) for 30 min on ice followed by centrifugation at 14,000×$g$ for 30 min to yield nuclear extract. Anti-FLAG agarose beads (Sigma-Aldrich) were incubated in the nuclear extract at 4°C for 2 hr and washed with nuclear extraction buffer. The beads were eluted with 0.1 M glycine-HCl pH 3.5, and the neutralized eluate was analyzed by Western blotting with the indicated antibodies.

## P-TEFb kinase assays

Kinase assays were performed in LoBind tubes (Eppendorf) in 20 µl reactions containing 50 mM HEPES pH 7.3, 50 mM NaCl, 1 mM DTT, 10 mM $MgCl_2$, and 0.05 mM or 0.5 mM ATP. For assays with 0.05 mM ATP, 0.15 µCi [$^{32}$P]-γ-ATP was used, and for reactions with 0.5 mM ATP, 1.5 µCi [$^{32}$P]-γ-ATP was used. Reactions in conventional tubes gave distinct less reproducible results. Purified recombinant P-TEFb or P-TEFb-Tat was pre-incubated with 500 ng purified recombinant GST-CTD (52 C-terminal domain repeats from human RNA polymerase) in the absence or presence of 0.3 µM purified AFF4$_{2-73}$ for 15 min at 20°C. After addition of ATP, the kinase reactions were stopped at different times by addition of 5 µl of 5× SDS sample buffer. The samples were analyzed by SDS-polyacrylamide gel electrophoresis, followed by measurement of the radioactive protein bands on a Typhoon phosphorimager (GE Healthcare).

For Western blots of kinase reactions performed with non-radioactive ATP, 0.2 µg of GST-CTD from each reaction was loaded per lane of a 4–20% SDS-PAGE gel, transferred onto a PVDF membrane (Immobilon-FL; Millipore, Billerica, MA) and processed by standard Western Blot procedures. The primary antibodies were ab5095 (αpS2; Abcam) and ab5131 (αpS5; Abcam) at 1:1000 dilution for both. The secondary antibody was a fluorescently labeled goat anti-rabbit antibody (Odyssey; LI-COR Biosciences, Lincoln, NE) at 1:20,000 dilution.

## Acknowledgements

Ann Fischer helped with baculovirus production, David King performed mass spectrometry, and Scott Endicott synthesized and purified peptides. We are grateful to James Holton, George Meigs, and Jane Tanamachi at Beamline 8.3.1 at Lawrence Berkeley National Laboratory for help with X-ray data collection. We appreciate the encouragement of the HARC Center.

## Additional information

### Funding

| Funder | Grant reference number | Author |
| --- | --- | --- |
| National Institutes of Health | R01AI41757-11, R01AI095057 | Qiang Zhou |
| California HIV/AIDS Research Program | ID09-B-026 | Tom Alber |
| National Institutes of Health | P50GM82250 | Tom Alber, Nevan J Krogan |

The funders had no role in study design, data collection and interpretation, or the decision to submit the work for publication.

### Author contributions

US-G, HU, AB, Conception and design, Acquisition of data, Analysis and interpretation of data, Drafting or revising the article; KB, SC, Conception and design, Acquisition of data; NJK, QZ, TA, Conception and design, Analysis and interpretation of data, Drafting or revising the article

## Additional files

### Major datasets

The following dataset was generated:

| Author(s) | Year | Dataset title | Dataset ID and/or URL | Database, license, and accessibility information |
|---|---|---|---|---|
| Alber T, Schulze-Gahmen U | 2013 | Data From: The AFF4 scaffold binds human P-TEFb adjacent to HIV Tat | 4IMY; http://www.rcsb.org/pdb/search/structidSearch.do?structureId=4IMY | Publicly available at the RCSB Protein Data Bank (http://www.rcsb.org/pdb/). |

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
