## [Decision Letter]

Thank you for choosing to send your work entitled “The aff4 scaffold binds human P-TEFb adjacent to HIV Tat” for consideration at *eLife*. Your article has been evaluated by a Senior editor and 2 reviewers, one of whom is a member of our Board of Reviewing Editors.

The following individuals responsible for the peer review of your submission want to reveal their identity: Wes Sundquist, Reviewing editor. The Reviewing editor and the other reviewer discussed their comments before we reached this decision, and the Reviewing editor has assembled the following comments based on the reviewers' reports.

The authors describe the structure and associated biochemistry of the complex between P-TEFb and the relevant binding region of the SEC scaffold, AFF4. Overall, this work is of high technical quality and represents an important advance.

The crystal structure reveals that an otherwise unstructured epitope within aff4 (residues 34–66) binds P-TEFb, and meanders across an exposed surface of the Cyclin T1 subunit. The structure is at relatively low resolution (2.9 Å), but the crystallography appears technically strong and the overall similarity of the three complexes in the asymmetric unit also lends confidence. The structure reveals how P-TEFb binds the aff4 scaffold, and the observed Cyclin T1-AFF4 interactions are strongly supported by extensive binding studies using mutant Cyclin T1 proteins and transcription activation data using mutant aff4 proteins. A very interesting point is that the bound aff4 peptide approaches the Tat binding site, suggesting that aff4 may contact Tat directly. This hypothesis is nicely supported by the demonstration that aff4 binds 11-fold more tightly to the P-TEFb:Tat complex than to P-TEFb alone. Supporting IP data also show that Tat expression can “rescue” the effects of aff4 mutations that would otherwise inhibit Cyclin T1 binding, although these studies do not unambiguously identify aff4 residues that contact Tat (Glu61/Met62 are presented as attractive candidates, but the binding/rescue effects are small). The discovery that Tat probably contacts aff4 directly is significant, and adds aff4 as an important component to the previously characterized pTEFb/Tat/TAR complex.

The authors should provide:

1) Additional evidence for the modeled Tat-AFF4 interaction. This could be done by testing the effects of mutations at aff4 positions 61-63 on Tat enhancement of aff4 binding to P-TEFb in the in vitro binding assay. This would provide a clear quantitative assessment of the modeled binding pocket.

2) Additional analysis of the (putative) AFF4-CDK9 interaction. This contact is only seen in one of the complexes in the asymmetric unit, but multiple binding experiments suggest that interactions the N-terminal region of aff4 and the CDK9 subunit probably increase the affinity of the AFF4:P-TEFb complex by about four-fold (e.g., the data in Table 2), and mutations in this region of aff4 can have significant (albeit not overwhelming) effects on transcriptional activation (Figure 3B). Testing one or two structure-based aff4 point mutants would address whether or not this (admittedly weak) contact actually occurs in solution. Both types of requested aff4 mutants could be analyzed in the FP competition experiments (as reported for other aff4 constructs in Table 2).

---

## [Author Response]

We would like to explore with you the necessity of providing additional data to assess 1) the Tat interaction with aff4 and 2) the aff4 interaction with CDK9. In general, we are concerned that the suggested experiments would not be definitive and that they would delay dissemination of the discovery of the intersubunit pocket for Tat.

As appreciated in the review, we provided significant evidence for a direct Tat interaction with AFF4. In addition to the proximity of aff4 and Tat in the respective cocrystal structures, we showed that Tat enhances aff4 binding 11-fold in vitro, that Tat rescues P-TEFb binding to aff4 interface mutants in vivo, and that the aff4 61/62 double alanine substitution in the proposed Tat interface blocks this rescue. Moreover, K28 of Tat, which is engaged in the aff4 interaction in our model but exposed to solvent in the Tat-P-TEFb complex, is acetylated in vivo and this modification regulates Tat functions.

While each of these experiments is not by itself definitive, together they make the case for a direct Tat-AFF4 interaction. Indeed, the suggestion that we assay quadruple aff4 mutants including the 61/62 di-alanine substitution is predicated partly on our result that this double mutation blocks Tat rescue of the interaction of P-TEFb with the mutant AFF4. There is no doubt that the requested experiments could provide additional examples of the concepts we demonstrated, but they would not add new lines of proof.

Similarly, analysis of additional mutants in the N-terminal helix stabilized by intermolecular interactions in the crystals would not provide definitive evidence about the partner of this sequence. If we had proposed that this segment of aff4 functions to bind CDK9, we suspect the reviewers would have noted our naïveté about an interaction of a 20-residue helix between molecules in one of 3 crystal complexes that makes a non-complementary interface and contributes less than one kcal per mole to the affinity. Moreover, they would have pointed out that the 13/14 di-alanine mutant in the heart of the interface is associated with a <25% decrease in transcription stimulation (Figure 3B) and little change in P-TEFb binding (Figure 4C) in vivo.

Rather than making and assaying a handful of additional mutants, we edited the manuscript to clarify the conclusions and tie them back to the data.